# Convenience Food Options and Adequacy of Nutrient Intake among School Children during the COVID-19 Pandemic

**DOI:** 10.3390/nu14030630

**Published:** 2022-01-31

**Authors:** Nihaal Rahman, Kazue Ishitsuka, Aurélie Piedvache, Hisako Tanaka, Nobuko Murayama, Naho Morisaki

**Affiliations:** 1Department of Social Medicine, National Center for Child Health and Development, 2-10-1 Okura, Setagaya-ku, Tokyo 157-8535, Japan; nihaal-r@ncchd.go.jp (N.R.); aurelie-p@ncchd.go.jp (A.P.); tanaka-hs@ncchd.go.jp (H.T.); morisaki-n@ncchd.go.jp (N.M.); 2Department of Health and Nutrition, Faculty of Human Life Studies, University of Niigata Prefecture, 471 Ebigase, Higashi-ku, Niigata 950-8680, Japan; murayama@unii.ac.jp

**Keywords:** COVID-19, convenience food, take-out food, diet quality, school children, nutrition, Japan

## Abstract

The COVID-19 pandemic has caused changes in the family food environment, resulting in more families relying on convenience food options. This study aimed to investigate diet quality by convenience food options (namely instant, frozen, and take-out foods) among Japanese school children during the COVID-19 pandemic. We examined the relationship between the frequency of consumption of convenience food options and nutritional status of the school children. The participants (671 children, 10–14 years old) were chosen to form a nationally representative sample of the Japanese population. Using questionnaires completed by the participants’ guardians, information was collected on the frequency of instant, frozen, and take-out food consumption. Habitual food and nutrient intake were collected using a validated food frequency questionnaire, completed by the children with help from their guardian(s). “Frequent” consumption was defined as consumption of instant, frozen, and/or take-out foods on more than 5 days per week. Using 19 nutrients and their respective dietary reference intake (DRI) values, an index was created to label each child’s nutrient intake as “Adequate”, “Inadequate”, “Excess”, or “Deficient.” Compared to children with non-frequent consumption, school children with frequent instant food consumption had significantly higher rates of inadequate nutrient intake (risk ratio (RR) = 3.0 [95% CI: 1.6–5.6]) and excess nutrient intake (RR = 2.3 [95% CI: 1.3–4.2]), while school children with frequent take-out food consumption had significantly higher rates of inadequate nutrient intake (RR = 2.1 [95% CI: 1.3–3.3]). There were no significant differences for children with frequent frozen-food intake. These associations did not change when adjusting for sociodemographic factors. Our results suggest that the frequent consumption of instant or take-out foods among school children results in non-adequate nutritional intake.

## 1. Introduction

Since being declared a pandemic in March 2020 by the World Health Organization (WHO) [1], the coronavirus disease 2019 (COVID-19) caused by severe acute respiratory syndrome coronavirus 2 (SARS-CoV-2) continues to spread worldwide [2]. Since the beginning of the pandemic, there have been over 1.7 million confirmed cases of COVID-19 and over 18,000 COVID-19-related deaths in Japan [3]. In addition to the impact on human health, the COVID-19 pandemic affected a variety of lifestyle practices, including food behavior. Before the pandemic, Japan already had one of the higher country-wide retail sales of ultra-processed foods (such as packaged snacks or prepared frozen dishes), which is a strong indicator of unfavorable nutrient intake [4]. Once the pandemic began, countries around the world saw an increase in the purchases and consumption of convenience foods, such as instant and frozen foods [5]. Additionally, in Japan, those with worsened household income self-reported less time for cooking and higher rates of consuming take-out foods [6]. Therefore, these changing household food environments are expected to have a direct impact on the nutritional intake of school children.

Most studies analyzing the nutritional value of instant, frozen, and take-out foods focus primarily on adult populations. These studies have found that increased consumption of instant foods is associated with lower intakes of healthy foods, such as fruits and vegetables, and is subsequently associated with lower intake of essential nutrients [7]. Similarly, frequent take-out food consumption is associated with a lower micronutrient intake and higher fat intake [8]. However, adequate nutrition during childhood is vitally important for growth, development, and the prevention of chronic disease development in later life [9]. Micronutrient deficiencies during school ages could have severe consequences for physical and mental health [10]. With more school-aged children living and eating at home during the span of the COVID-19 pandemic, it is important to understand the effect of this increased convenience food consumption. Moreover, at the time of writing this paper, the pandemic is still ongoing; therefore, convenience food consumption may continue to increase as time goes on. Therefore, it is of particular importance to consider children’s nutrient intake during the COVID-19 pandemic in response to these food behavior changes.

In this study, we assessed the quality of Japanese school children’s diets during the COVID-19 pandemic, particularly in reference to the frequency at which school children were consuming instant, frozen, or take-out foods.

## 2. Materials and Methods

### 2.1. Study Participants

A stratified two-stage clustering design was used to obtain a nationally representative sample of households in Japan; the sampling frame used the resident registration system from the fiscal year 2020. All residents of Japan must be registered, according to the Residential Basic Book Act. We stratified based on the 8 regions of Japan: Hokkaido and Tohoku, Kanto, Hokuriku/koushin’etsu, Chubu, Kinki, Chugoku, Shikoku and Kyushu, and Okinawa. A random sample of 6–7 municipalities was drawn from each prefecture, producing a total of 50 municipalities in the first sampling stage. In the second sampling stage, 30 households with children in grade 5 (10–11 years old) or grade 8 (13–14 years old) were randomly selected from each municipality, resulting in a total of 3000 households extracted. Grades 5 and 8 were chosen as representative of higher elementary school and middle school with reference to previous studies on adolescents in Japan [11]. Half of the extracted households were additionally sent brief self-administered diet history questionnaires (BDHQs). Out of 1500 households administered the BDHQ, 765 (51.0%) completed the survey. Using the reported energy intakes of the 765 respondents, we defined a restricted range of plausible energy intakes to assess underreporting and overreporting. School children were excluded when their reported energy intake was below half of their respective estimated energy requirement (EER), or above 1.5 times their respective EER [12]. Excluding the outliers resulted in a final sample size of 671 participants. This study was conducted in accordance with the Declaration of Helsinki and the Ethical Guidelines for Clinical/Epidemiological Studies of the Japanese Ministry of Health, Labor, and Welfare. Ethical approval was obtained from the ethics committee of the National Center for Child Health, Japan (No. 2020-168) and the University of Niigata Prefecture, Japan (No. 2025). Written informed consent was obtained for all subjects.

### 2.2. BDHQ for Children and Adolescents

The BDHQ for children and adolescents was developed based on the BDHQ for the general population [13]. The BDHQ was developed as a faster way to assess dietary intake through the collection of information on 58 food items and has been validated through dietary records and various biomarker methods, allowing them to serve as useful tools in nutritional epidemiologic studies [14]. For this study, we used BDHQ responses from Japanese children and adolescents in grades 5 or 8 to investigate the association between the frequency of convenience food consumption and nutritional intake.

### 2.3. Convenience Foods

The questionnaire asked guardians for information on the frequency at which their child ate instant foods, frozen foods, and take-out foods. These three food-groups are generally considered to be convenience food options in its broadest sense [15]. Participants were asked to select consumption frequency from the following statements for each of the three types of convenience foods: (i) almost every day, (ii) 4–5 times a week, (iii) 2–3 times a week, or (iv) rarely. Based on these responses, we categorized the participants’ intake of the three convenience food groups as either being seldom (rarely) or frequent (almost every day, 4–5 times/week, 2–3 times/week).

### 2.4. Nutrient Adequacy

While food frequency questionnaires can estimate the composition of food intake, their ability to capture absolute intake is limited. Thus, to compare the dietary intakes reported in the survey with corresponding Dietary Reference Intake (DRI) values, we adjusted the reported nutrient intakes to energy-adjusted intake values [12]. We assumed each participant consumed the respective EER for age/sex, rather than the reported energy, to calculate the energy-adjusted values. Physical activity level was assumed to be level II (moderate) for all participants due to an absence in information regarding physical activity. The calculation is as follows: energy-adjusted nutrient intake (amount/day) = [reported nutrient intake (amount/day) × EER (kcal/day)]/[observed energy intake (kcal/day)]. Nineteen nutrients were assessed in this study, with protein being assessed under two different metrics. Protein, fat, carbohydrates, dietary fiber, sodium, and potassium intake values were compared to their respective Japanese DRI tentative dietary goal for preventing lifestyle-related diseases (DG) values. Protein, vitamin A, thiamin, riboflavin, niacin, vitamin B6, vitamin B12, folate, vitamin C, calcium, magnesium, iron, zinc, and copper intake values were compared to their respective Estimated Average Requirement (EAR) values.

In addition to an analysis on each nutrient, a DRI index for diet adequacy was conducted using the 19 nutrients [16]. For this, the total adequacy of nutrient intake was categorized into four groups based on the number of nutrients that met the DG (maximum = 6) and EAR (maximum = 14). Therefore, finally, the indexing labels were defined as follows: ‘Adequate’ (EAR ≥ 12, DG ≥ 4), ‘Excess’ (EAR ≥ 12, DG ≤ 3), ‘Deficient’ (EAR ≤ 11, DG ≥ 4), or ‘Inadequate’ (EAR ≤ 11, DG ≤ 3).

### 2.5. Demographics

The questionnaire asked for information on gender, age (categorized into preteen (10/11 years old) and teen (13/14 years old)), family size (2, 3, 4, 5, or 6 and more), household income, economic situation during COVID-19 pandemic compared to December 2019 (worse than before, no different, better than before, or do not want to answer), education attainment level of the guardians (less than high school, high school, vocational school, junior college, university or graduate school, or no biological parent or does not want to answer), and region. Body mass index (BMI) was calculated through self-reported height and body weight of each child. BMI percentiles are the recommended values from the Japanese Society of Pediatric Endocrinology and the Japanese Association for Human Auxology [17,18].

Additionally, the survey we administered included seven questions related to meal preparation literacy. Three questions assessed the guardian’s ability (skill) to prepare meals for their children, two questions assessed their knowledge on what a healthy diet is, and two questions assessed their attitudes towards providing healthy meals to their children. Each response was rated with a 5-point Likert scale (1, yes/always; 2, yes/sometimes; 3, unsure; 4, no/not so much; and 5, no/not at all for skill and attitude questions: 1, not at all; 2, understand a little; 3, neither; 4, understand mostly; 5, fully understand for knowledge questions). These variables were treated as numerical, and the scores for attitude and affordability questions were reversed. Then, the final score for each food literacy group was obtained from adding their respective questions. The maximum score for each subgroup is as follows: 15 for knowledge (3 questions, maximum score of 5 per question), and 10 for skill and attitude (2 questions each, maximum score of 5 per question).

### 2.6. Statistical Analysis

Categorical variables were expressed as percentage of participants; numerical variables were expressed as median and interquartile range (IQR). The χ^2^ test was used to compare categorical data between frequent and seldom consumers of each food group type, a Fisher test was used for observations under 5, and a Wilcoxon rank sum test was used for numerical variables. Linear regressions were conducted to check for confounding when discussing food and nutrient intake. The association between convenience food intake and nutrient adequacy was assessed with a multinomial logistic regression conducted for each of the three convenience food options. We performed crude and adjusted regressions. Adjustment used gender and grade plus one additional variable (1, family size; 2, father’s education level; 3, mother’s education level; 4, household income; 5 the literacy score for skill; or 6, the literacy score for attitude) independently because of possible multicollinearity between these six variables. Relative risks (RR) using an “adequate” nutrient intake as reference are reported for “excess”, “deficient”, and “inadequate” diets. All *p*-values were two-sided, and the significance level was <0.05. All statistical analyses were performed using R Version 1.2.5042.

## 3. Results

Table 1 lists the sociodemographic characteristics of the participants. Among those with instant food intake information, significant differences were observed between frequent and seldom consumers for gender, family size, household income, and educational attainment level of the father. Among those with frozen food intake information, significant differences were only observed for grade-level. Among those with take-out food intake information, significant differences were observed for the household income and educational attainment level of both the mother and father.

Table 2 lists the means and standard deviations of the skill, knowledge, and attitude literacy scores of the participants based on frequency of convenience food consumption. For those with instant food intake information, there were significant differences in the skill and attitude scores between frequent and seldom consumers. For those with frozen food intake information, there was a significant difference in the skill scores between frequent and seldom consumers. Finally, for those with take-out food intake information, there was a significant difference in the attitude scores between frequent and seldom consumers.

Table 3 lists the median and IQR values for 33 food and nutrient items from the BDHQ survey. Between frequent and seldom consumers, for instant food intake, there were significant differences in 16 of the items listed, with strong differences (*p*-values < 0.001) in “fruits and vegetables” and soda for food items, and vitamin A, folate, calcium, iron, magnesium, phosphorus, and potassium for nutrients. A notable difference in the median intakes of “meat and eggs” could be observed, although the difference was insignificant. For frozen food, the intake of six of the items listed showed significant differences; however, no food items or nutrients showed strong differences. A notable difference in fruits and vegetables consumption could be observed, although the difference was insignificant. Finally, between frequent and seldom consumers of take-out food, there were significant differences in 14 of the items listed, with strong differences in soda for food items and dietary fiber, magnesium, phosphorus, and potassium for nutrients. Insignificant but notable differences in “meat and eggs” consumption were also observed.

Additionally, the results from the linear regression models on gender, age, and income are shown in this table. For instant food intake, 14 of the 16 food and nutrient items with significant differences were still found to have significant differences after regression modeling, with saturated fatty acids and protein no longer demonstrating significant differences. For frozen food intake, three food and nutrient items were found to have significant differences after regression modeling; however, only magnesium was consistent between the crude and regression analyses. Between frequent and seldom frozen food consumers, “meat & eggs” and “nuts & pulses” consumption display significant differences only after regression analysis. Finally, for take-out food intake, 14 of the items listed display strong differences, the difference being that vitamin A intake no longer displays significance after regression modeling, while a significant difference was observed for confectionary consumption only after regression modeling.

The comparisons to standard intake recommendations and subsequent dietary indexing are listed in Table 4. Among those with instant food intake information (*n* = 666), there were significant differences in nine of the twenty nutritional metrics listed by frequency of instant food intake (all *p* < 0.05), namely protein, dietary fiber, vitamin A, thiamin, folate, vitamin C, potassium, calcium, and magnesium. Those who consume instant foods frequently had a significantly greater chance of having a non-adequate diet compared to the seldom consumers (*p* = 0.001). For those with frozen food information (*n* = 660), there were significant differences in four of the metrics, namely protein, dietary fiber, thiamin, and calcium. No significant difference was observed in non-adequate diets between frequent and seldom consumers of frozen foods. Among those with take-out food information (*n* = 663), there were significant differences in seven of the metrics (all *p* < 0.05), namely protein, dietary fiber, vitamin A, thiamin, potassium, calcium, and magnesium. Those who consume take-out foods frequently had a significantly greater chance of having a non-adequate diet (*p* = 0.006).

Table 5 shows the results of the crude and multinomial logistic regression models and calculated RRs with 95% Cis. Among those with instant food intake information, a significantly greater risk for excess or inadequate diets was observed. Among those with frozen food intake information, there was no significantly greater risk for a non-adequate diet observed. Among those with take-out food intake information, there was an observed greater risk of inadequate diets. Adjusting for gender, grade, and income did not change these associations. Additional regression models for sociodemographic factors and a comparison between those with and without income data can be found in Appendix A, respectively. 

## 4. Discussion

The COVID-19 pandemic continues to severely impact communities around the world [2]. With lifestyles and livelihoods being affected by the pandemic, there has been an increased reliance on convenience food options during the span of the pandemic [6]. We found that such changing family food environments, especially an increase in instant food and take-out food consumption, have the potential to affect the nutritional intake of children. Children identified as having frequent instant food consumption were found to have significantly higher rates of inadequate nutrient intake and excess nutrient intake. They had significant differences in the consumption of 16 food and nutrient items, with strong differences in “fruits and vegetables,” soda, vitamin A, folate, calcium, iron, magnesium, phosphorus, and potassium. Those with frequent take-out food consumption had significantly higher rates of inadequate nutrient intake. They also had significant differences in 14 food and nutrient items, with strong differences in soda, dietary fiber, magnesium, phosphorus, and potassium. While those with frequent frozen food intake had no significant differences between adequate and non-adequate nutrient intakes, there were significant differences in six food and nutrient items, albeit not strong differences.

The associations between convenience food intake and non-adequate nutritional status have yet to be extensively studied in Japan. However, in a study of Korean adults, those with frequent instant noodle consumption were shown to have significant differences in the intakes of nuts, fruits, vegetables, protein, calcium, phosphorus, iron, potassium, vitamin A, and vitamin C [7]. Moreover, in a study of Korean children and adolescents, instant noodle consumption was associated with significant differences in the intake of fruits, vegetables, calcium, and vitamin C [19]. These food groups and nutrient deficiencies based on instant food consumption are all supported by the findings of our own study. However, in both these studies, the researchers found a significantly lower intake of potatoes in the instant-food group, whereas, in our study, we found a significantly higher intake of potatoes in the instant-food group. Additionally, the study in Korean adults found no significant difference in dairy consumption and a significant difference in meat, eggs, and fish consumption, whereas we found a significant difference in dairy and no significant difference in meat, eggs, and fish. Due to the different study populations, these differences may be the result of different food environments between Japan and Korea [20].

The associations between frequent frozen or take-out food consumption and nutritional status are even less studied than those of instant foods. One study conducted in Australia found that “less-healthy” take-out food consumption partially explained lower fruit and vegetable intake among those of a lower socioeconomic status (SES) [21]. This supports our own findings as we found a significant difference (*p* < 0.05) in fruit and vegetable consumption between frequent and seldom take-out food consumers. While we found take-out food consumption is associated with poor nutrient intake, there is a lack of studies investigating this association. Further research is needed on the nutritional status of frequent take-out or frozen food consumers compared to seldom consumers.

There have been studies conducted across multiple countries examining the relationship between SES and dietary intake [22]. In Japanese adults, a lower SES was found to be associated with a lower quality of food and nutrient intake [23]. Moreover, children living in lower-income households in Japan were found to have a lower frequency of “well-balanced” meals since the COVID-19 state of emergency [24]. In our study, we found correlations between instant food and take-out food consumption and lower SES, with resulting implications for non-adequate nutritional intake. While we do not know if convenience food intakes have increased overall, it is possible that this is having an impact on a subpopulation in Japan. Further studies are required to see which groups are purchasing and/or have purchased convenience food items during the span of the COVID-19 pandemic, and the extent to which those purchasing patterns have increased pre-pandemic.

At the start of the COVID-19 pandemic, it was predicted that malnutrition among children was likely to increase due to the impacts the pandemic had on poverty, coverage of essential interventions, and access to nutritious foods [25]. The social effects caused by the pandemic have also been found to lead to undernutrition or obesity, such as isolation and physical inactivity [26]. In Japan, those who were living alone or had higher stress levels were seen to change their diet habits unhealthily during the COVID-19 pandemic [27]. Moreover, those whose household income decreased or whose economic status worsened were found to decrease the time and effort for cooking and increase take-out food consumption [5]. Before the pandemic, habitual inadequacies were found in fiber, sodium, and fat intake [16]. We found these same inadequacies in our own study but also found greater inadequacies of additional nutrients, such as thiamin and carbohydrates. Thus, our results suggest that the COVID-19 pandemic has worsened the status of nutritional intake among Japanese school children compared to before the pandemic.

While BMI was higher in the frequent consumer group across all three convenience food groups, we failed to observe significant differences between frequent and seldom consumers. There might be a few reasons why we did not observe this result in our study. Other studies on the consumption of ultra-processed food showed large differences in BMI/obesity [28]. While this association may be present in our study, we may not have had the power to observe it. This could also potentially be explained by the school lunch program in Japan as it was designed to provide Japanese school children with identical healthy meals and has been proven to decrease the percentage of overweight and obesity among boys [29]. During the time of our study, school children were attending school; therefore, they had access to these school lunches. These school lunches provide some degree of a preventive effect against overweight and obesity, thereby making BMI a less sensitive measure in our study. On the other hand, our results are consistent with the studies on Korean adults and children and their respective instant noodle consumption [7,19]. While instant and frozen foods are examples of ultra-processed foods, these results suggest that the correlation between the intake of convenience foods and obesity may be weaker than that of ultra-processed foods, generally, and obesity. More studies are needed to better understand the association between convenience foods and BMI.

The nutritional deficiencies noted in this study have serious implications for the health of school children. Epidemiologic studies find that greater consumption of fruits and vegetables is associated with a reduced risk of mortality from all causes, and particularly cardiovascular disease [30]. Moreover, consumption of sugary drinks, such as soda, is associated with an increased risk of being overweight or obese, diabetes, hypertension, and cancer [31]. This means children with frequent instant food consumption are at risk of increasing their risk for these adverse health effects, while they are also losing out on the preventative effects associated with frequent fruit and vegetable consumption. Those with frequent instant food intake are also at risk of essential nutrient deficiencies. Vitamin A deficiency has been associated with a variety of clinical diseases, ranging from xerophthalmia to reduced resistance to infection [32]. Folate deficiencies have been associated with an increased risk in chronic disease development, such as several cancers, neurological disease, and neurologic conditions [33]. Both frequent instant food and take-out food consumers demonstrated significantly lower intakes of potassium and magnesium, deficiencies of which are associated with the risk of various chronic diseases, such as hypertension and neuromuscular irritability, respectively [34,35]. While there is also an observed significant difference in phosphorus intake, phosphorus deficiency is rare in the healthy population [36]. Finally, dietary fiber intake is associated with a lowered risk of total death from cardiovascular disease and all cancers [37]. Therefore, both the frequent instant and take-out food groups demonstrate a greater risk of chronic disease development while also failing to benefit from the preventative effects of certain foods and nutrients.

The present study benefitted from being among the first studies to investigate the nutritional intake of school children during the COVID-19 pandemic and being the first to specifically investigate the effects and associations of frequent convenience food consumption. Moreover, another strength of this study was the use of a nationally representative survey to enroll participants. However, this study also had several limitations. Firstly, a moderate response rate of 51.0% may limit the generalizability of our findings. Nevertheless, the characteristics of the school children in our study are comparable to those in the Japanese national survey. Moreover, the household income in the present study was comparable to those among households with children in the Japanese national survey [24,38]. Secondly, the potential for inaccuracies exists due to BDHQ self-reporting, such as over- or underestimating. While the BDHQ serves as a useful tool for diet recall over a longer period, the additional use of measurements biomarkers would strengthen the analysis of such a study. Thirdly, this study cannot be generalized to an older population or a non-school attending population due to the inclusion of only grade-school aged children and the implications of school children having access to healthy school lunches that non-school children would lack. Finally, due to the cross-sectional nature of our study, our analyses were limited to reporting nutritional intake by convenience food option and we were not able to assess its long-term health effects. Future studies assessing the resulting potential for adverse health effects of altered nutritional intake due to increased convenience food intake are needed.

## 5. Conclusions

The evidence from our study suggests that frequent instant and take-out food consumption are associated with non-adequate nutritional intake. With the COVID-19 pandemic ongoing, there are likely to be even more children dependent on convenience food options in the near future. Providing convenience food options that are more nutrient-rich, as well as recommending families to choose healthier items when consuming convenience food options, may help to prevent long-term adverse health effects and are needed for children for whom a reduction in the intake of convenience food options itself is not possible.

## Figures and Tables

**Table 1 nutrients-14-00630-t001:** Sociodemographic factors of participants.

		Convenience Foods														
		Instant Food					Frozen Food					Take-Out Food				
		Frequent		Seldom			Frequent		Seldom			Frequent		Seldom		
		N = 133		N = 533			N = 230		N = 430			N = 271		N = 392		
		N	%	N	%	*p*	N	%	N	%	*p*	N	%	N	%	*p*
Gender	Boys	78	58.6	244	45.8	0.010	108	47.0	210	48.8	0.705	137	50.6	184	46.9	0.403
	Girls	55	41.4	289	54.2		122	53.0	220	51.2		134	49.4	208	53.1	
Grade	Preteen	63	47.4	284	53.3	0.261	105	45.7	238	55.3	0.022	130	48.0	216	55.1	0.084
	Teen	70	52.6	249	46.7		125	54.3	192	44.7		141	52.0	176	44.9	
Family size	2	7	5.3	25	4.7	0.016	8	3.5	24	5.6	0.498	12	4.4	20	5.1	0.542
	3	19	14.3	74	13.9		31	13.5	61	14.2		40	14.8	53	13.5	
	4	41	30.8	237	44.5		100	43.5	176	40.9		121	44.6	157	40.1	
	5	38	28.6	131	24.6		65	28.3	103	24.0		59	21.8	107	27.3	
	6+	27	20.3	60	11.3		26	11.3	59	13.7		36	13.3	51	13.0	
	No response	1	0.8	6	1.1		0	0.0	7	1.6		3	1.1	4	1.0	
Household annual income (JPY)	Under 2 million	9	6.8	24	4.5	<0.001	9	3.9	24	5.6	0.712	22	8.1	11	2.8	0.010
	2–4 million	31	23.3	56	10.5		35	15.2	52	12.1		39	14.4	47	12.0	
	4–6 million	31	23.3	118	22.1		53	23.0	94	21.9		66	24.4	81	20.7	
	6–8 million	29	21.8	112	21.0		44	19.1	95	22.1		57	21.0	83	21.2	
	8–10 million	9	6.8	80	15.0		31	13.5	58	13.5		27	10.0	62	15.8	
	Over 10 million	7	5.3	68	12.8		28	12.2	46	10.7		28	10.3	48	12.2	
	No response	17	12.8	75	14.1		30	13.0	61	14.2		32	11.8	60	15.3	
Economic circumstance after COVID-19	Worse than before	49	36.8	136	25.5	0.076	66	28.7	117	27.2	0.508	85	31.4	97	24.7	0.224
	No different	79	59.4	368	69.0		156	67.8	287	66.7		173	63.8	274	69.9	
	Better than before	4	3.0	20	3.8		5	2.2	19	4.4		8	3.0	16	4.1	
	Do not want to answer	1	0.8	9	1.7		3	1.3	7	1.6		5	1.8	5	1.3	
BMI	<10%	16	12.0	58	10.9	0.493	25	10.9	49	11.4	0.981	34	12.5	40	10.2	0.560
	10–90%	94	70.7	391	73.4		168	73.0	313	72.8		191	70.5	292	74.5	
	>90%	11	8.3	30	5.6		14	6.1	26	6.0		17	6.3	23	5.9	
	No response	12	9.0	54	10.1		23	10.0	42	9.8		29	10.7	37	9.4	
Education level of mother	Less than high school	4	3.0	13	2.4	0.209	5	2.2	11	2.6	0.916	9	3.3	8	2.0	0.027
	High school	42	31.6	129	24.2		58	25.2	113	26.3		83	30.6	86	21.9	
	Vocational	31	23.3	114	21.4		52	22.6	92	21.4		63	23.2	81	20.7	
	Junior college	33	24.8	131	24.6		52	22.6	110	25.6		59	21.8	104	26.5	
	University/graduate school	22	16.5	138	25.9		60	26.1	98	22.8		53	19.6	108	27.6	
	No biological mother/No response	1	0.8	8	1.5		3	1.3	6	1.4		4	1.5	5	1.3	
Education level of father	Less than high school	11	8.3	18	3.4	0.002	7	3.0	21	4.9	0.521	10	3.7	18	4.6	0.040
	High school	51	38.3	161	30.2		82	35.7	128	29.8		104	38.4	105	26.8	
	Vocational	19	14.3	67	12.6		25	10.9	61	14.2		34	12.5	52	13.3	
	Junior college	4	3.0	11	2.1		5	2.2	10	2.3		5	1.8	10	2.6	
	University/graduate school	38	28.6	249	46.7		99	43.0	185	43.0		102	37.6	186	47.4	
	No biological father/No response	10	7.5	27	5.1		12	5.2	25	5.8		16	5.9	21	5.4	
Region	Chugoku	19	14.3	70	13.1	0.236	35	15.2	52	12.1	0.755	38	14.0	51	13.0	0.915
	Chubu	19	14.3	68	12.8		28	12.2	58	13.5		30	11.1	56	14.3	
	Kyushu and Okinawa	22	16.5	61	11.4		30	13.0	51	11.9		30	11.1	51	13.0	
	Hokkaido and Tohoku	18	13.5	54	10.1		27	11.7	45	10.5		31	11.4	41	10.5	
	Hokuriku koushinetsu	10	7.5	66	12.4		27	11.7	48	11.2		33	12.2	43	11.0	
	Shikoku	14	10.5	61	11.4		20	8.7	55	12.8		32	11.8	43	11.0	
	Kinki	20	15.0	75	14.1		31	13.5	64	14.9		41	15.1	53	13.5	
	Kanto	11	8.3	78	14.6		32	13.9	57	13.3		36	13.3	54	13.8	

BMI: Body mass index.

**Table 2 nutrients-14-00630-t002:** Food literacy by convenience food consumption.

	Food Group														
	Instant Food					Frozen Food					Take-Out Food				
	Frequent		Seldom			Frequent		Seldom			Frequent		Seldom		
	N = 133		N = 533			N = 230		N = 430			N = 271		N = 392		
	Mean	SD	Mean	SD	*p*	Mean	SD	Mean	SD	*p*	Mean	SD	Mean	SD	*p*
Skill score (max = 15)	8.1	3.0	9.8	3.3	<0.001	8.9	3.4	9.8	3.2	0.002	8.9	3.3	9.9	3.3	0.058
Knowledge score (max = 10)	7.2	2.1	7.7	2.0	0.067	7.5	2.1	7.7	2.0	0.054	7.6	2.0	7.7	2.0	0.204
Attitude score (max = 10)	4.8	0.9	5.1	1.2	0.001	5.0	1.0	5.1	1.2	0.493	4.9	0.9	5.2	1.3	0.047

**Table 3 nutrients-14-00630-t003:** Food and nutrient intake by frequency of convenience food intake.

		Food Group											
		Instant Food				Frozen Food				Take-Out Food			
		Frequent	Seldom			Frequent	Seldom			Frequent	Seldom		
		N = 133	N = 533			N = 230	N = 430			N = 271	N = 392		
		Median (IQR)	Median (IQR)	*p*	*p**	Median (IQR)	Median (IQR)	*p*	*p**	Median (IQR)	Median (IQR)	*p*	*p**
Foods	Cereals (g/1000kcal)	206.7 (82.6)	201.2 (82.9)			203.9 (84.8)	201.1 (82.9)			204.8 (84.3)	200.9 (81.7)		
	Potatoes (g/1000kcal)	17.8 (16.8)	13.7 (14.2)	**	**	16.7 (15.5)	13.7 (14.2)	*		14.5 (14.6)	14.9 (15.3)		
	Sugar and sweeteners (g/1000kcal)	1.0 (1.0)	1.4 (1.2)	**	**	1.2 (1.1)	1.4 (1.2)			1.2 (1.1)	1.4 (1.2)	*	*
	Nuts and pulses (g/1000kcal)	16.6 (21.3)	20.7 (21.6)			19.3 (20.4)	20.1 (23.2)		*	18.2 (21.0)	20.8 (23.4)	*	*
	Fish (g/1000kcal)	24.7 (18.0)	25.0 (20.4)			24.6 (17.6)	25.2 (20.8)			24.6 (20.9)	25.1 (19.3)		
	Dairy products (g/1000kcal)	120.2 (114.8)	140.9 (128.9)	**	*	128.2 (127.4)	140.7 (122.7)			132.1 (135.9)	139.4 (119.8)		
	Oils (g/1000kcal)	6.5 (3.3)	6.5 (3.3)			6.6 (3.0)	6.6 (3.4)			6.5 (3.1)	6.5 (3.4)		
	Confectionaries (g/1000kcal)	23.8 (22.5)	25.3 (23.9)			25.6 (23.8)	24.8 (23.4)			26.6 (25.7)	23.6 (22.6)		*
	Seasonings (g/1000kcal)	123.6 (80.1)	123.0 (68.8)			123.2 (58.6)	123.5 (79.0)			121.7 (62.6)	123.7 (75.9)		
	Fruits and vegetables (g/1000kcal)	131.5 (73.4)	157.9 (91.5)	***	**	147.7 (94.0)	156.7 (93.0)			142.9 (85.9)	159.8 (92.1)	**	**
	Teas (g/1000kcal)	173.9 (197.2)	192.1 (229.5)			183.1 (201.1)	191.1 (233.1)			174.7 (203.8)	201.6 (229.6)		
	Meats and eggs (g/1000kcal)	53.3 (28.3)	57.3 (28.1)			53.7 (26.4)	57.5 (29.5)		*	55.4 (29.2)	56.6 (27.4)		
	Fruit and vegetable juice (g/1000kcal)	6.3 (17.9)	6.7 (17.2)			7.2 (26.4)	6.3 (16.1)			6.9 (25.2)	6.2 (15.1)		
	Soda (g/1000kcal)	31.4 (53.9)	10.2 (34.4)	***	**	14.7 (45.1)	10.6 (37.3)	*		19.1 (47.3)	9.6 (32.2)	***	**
Nutrients	Protein (g/1000kcal)	34.1 (7.3)	36.3 (8.0)	**		35.4 (7.5)	36.4 (8.0)	**		35.6 (8.3)	36.3 (7.7)	*	*
	Dietary fiber (g/1000kcal)	5.2 (1.5)	5.5 (1.9)	**	*	5.2 (1.5)	5.5 (2.0)			5.2 (1.6)	5.6 (1.9)	***	***
	n-3 polyunsaturated fat (g/1000kcal)	1.2 (0.4)	1.2 (0.4)			1.2 (0.4)	1.2 (0.4)			1.2 (0.4)	1.2 (0.4)		
	Saturated fatty acid (g/1000kcal)	11.1 (3.7)	11.6 (3.8)	*		11.6 (3.9)	11.5 (3.8)			11.5 (4.0)	11.6 (3.7)		
	Vitamin A (µg/1000kcal)	254.5 (136.5)	315.9 (174.1)	***	**	288.1 (174.4)	315.6 (167.1)			287.9 (181.8)	314.4 (159.7)	*	
	Folate (µg/1000kcal)	140.7 (53.4)	158.6 (62.5)	***	**	152.3 (55.0)	156.8 (63.4)			148.7 (58.1)	158.8 (65.2)	**	**
	Vitamin C (mg/1000kcal)	50.0 (22.1)	53.4 (25.8)	**	*	52.5 (24.9)	52.1 (25.4)			51.0 (23.5)	53.9 (26.9)	*	*
	Vitamin D (µg/1000kcal)	4.4 (2.7)	4.6 (3.1)			4.4 (2.8)	4.6 (3.0)			4.4 (3.0)	4.6 (2.9)		
	Calcium (mg/1000kcal)	323.9 (130.6)	365.0 (166.0)	***	**	336.7 (177.2)	367.1 (160.7)	*		338.4 (163.4)	370.3 (160.1)	**	*
	Iron (mg/1000kcal)	3.5 (0.8)	3.8 (1.1)	***	*	3.6 (0.8)	3.7 (1.2)			3.6 (1.0)	3.8 (1.1)	*	**
	Magnesium (mg/1000kcal)	112.6 (23.1)	123.5 (28.6)	***	***	119.8 (24.4)	123.1 (29.9)	*	*	118.4 (26.8)	124.1 (28.8)	***	***
	Phosphorus (mg/1000kcal)	550.2 (136.3)	590.3 (149.5)	***	**	569.9 (147.6)	591.2 (147.7)	*		567.9 (155.6)	591.4 (156.4)	***	**
	Potassium (mg/1000kcal)	1105.5 (285.7)	1221.2 (355.2)	***	**	1163.4 (335.2)	1223.2 (357.6)			1167.7 (329.8)	1225.8 (359.2)	***	***
	Sodium (mg/1000kcal)	5.5 (1.3)	5.4 (1.4)			5.4 (1.3)	5.5 (1.4)			5.4 (1.3)	5.5 (1.4)		

IQR: interquartile range. *p*-values were derived from a Wilcoxon rank sum test: * *p* < 0.05, ** *p* < 0.01, *** *p* < 0.001. *p** are *p*-values from linear regression model using gender, grade, and income: * *p* < 0.05, ** *p* < 0.01, *** *p* < 0.001.

**Table 4 nutrients-14-00630-t004:** Dietary reference index (DRI) comparison and dietary indexing.

		Food Group														
		Instant Food					Frozen Food					Take-Out Food				
		Frequent		Seldom			Frequent		Seldom			Frequent		Seldom		
		N = 133		N = 533			N = 230		N = 430			N = 271		N = 392		
		N	%	N	%	*p*	N	%	N	%	*p*	N	%	N	%	*p*
Nutrients	Protein (EAR)	133	100.0	533	100.0	N/A	230	100.0	430	100.0	N/A	271	100.0	392	100.0	N/A
	Protein (DG)	83	62.4	387	72.6	0.028	149	64.8	315	73.3	0.029	175	64.6	292	74.5	0.008
	Fat	53	39.8	209	39.2	0.972	85	37.0	175	40.7	0.393	116	42.8	145	37.0	0.154
	Carbohydrate	85	63.9	347	65.1	0.876	149	64.8	277	64.4	0.994	175	64.6	254	64.8	1.000
	Dietary fiber	23	17.3	159	29.8	0.005	50	21.7	131	30.5	0.021	55	20.3	127	32.4	0.001
	Vitamin A	99	74.4	463	86.9	0.001	188	81.7	369	85.8	0.207	218	80.4	342	87.2	0.023
	Thiamin	31	23.3	194	36.4	0.006	65	28.3	158	36.7	0.035	74	27.3	151	38.5	0.004
	Riboflavin	120	90.2	497	93.2	0.314	208	90.4	403	93.7	0.168	247	91.1	367	93.6	0.295
	Niacin	128	96.2	499	93.6	0.345	217	94.3	404	94.0	0.975	252	93.0	372	94.9	0.390
	Vitamin B6	107	80.5	453	85.0	0.251	190	82.6	364	84.7	0.569	220	81.2	337	86.0	0.122
	Vitamin B12	133	100.0	529	99.2	0.708	228	99.1	428	99.5	0.911	269	99.3	390	99.5	1.000
	Folate	127	95.5	527	98.9	0.024	227	98.7	421	97.9	0.677	263	97.0	388	99.0	0.124
	Vitamin C	113	85.0	487	91.4	0.040	207	90.0	388	90.2	1.000	241	88.9	357	91.1	0.436
	Salt	1	0.8	1	0.2	0.859	0	0.0	2	0.5	0.770	2	0.7	0	0.0	0.326
	Potassium	91	68.4	436	81.8	0.001	180	78.3	341	79.3	0.832	197	72.7	328	83.7	0.001
	Calcium	82	61.7	402	75.4	0.002	153	66.5	325	75.6	0.017	176	64.9	307	78.3	<0.001
	Magnesium	116	87.2	501	94.0	0.013	209	90.9	402	93.5	0.286	241	88.9	373	95.2	0.004
	Iron	95	71.4	416	78.0	0.133	180	78.3	326	75.8	0.541	199	73.4	308	78.6	0.150
	Zinc	131	98.5	531	99.6	0.379	228	99.1	428	99.5	0.911	268	98.9	391	99.7	0.378
	Copper	133	100.0	533	100.0	N/A	230	100.0	430	100.0	N/A	271	100.0	392	100.0	N/A
Index	Inadequate	44	33.1	117	22.0	0.001	61	26.5	100	23.3	0.517	84	31.0	77	19.6	0.005
	Adequate	18	13.5	162	30.4		54	23.5	123	28.6		63	23.2	117	29.8	
	Excess	68	51.1	244	45.8		110	47.8	199	46.3		121	44.6	189	48.2	
	Deficient	3	2.3	10	1.9		5	2.2	8	1.9		3	1.1	9	2.3	

EAR: Estimated Average Requirement. DG: Japanese DRI tentative dietary goal for preventing lifestyle-related diseases.

**Table 5 nutrients-14-00630-t005:** Crude and multinomial regressions for non-adequate diets.

		Relative Risk (CI 95%)			
		Adequate	Excess	Deficient	Inadequate
Instant Food (*n* = 666)	Crude (excluding missing income)	1.00	2.3 (1.3, 4.2) **	2.7 (0.7, 11.1)	3.0 (1.6, 5.6) **
	Adjusted (gender + grade + income)	1.00	2.2 (1.2, 4.0) **	3.0 (0.7, 12.5)	2.4 (1.2, 4.5) **
Frozen Food (*n* = 660)	Crude (excluding missing income)	1.00	1.3 (0.9, 2.0)	1.6 (0.5, 5.4)	1.4 (0.8, 2.2)
	Adjusted (gender + grade + income)	1.00	1.3 (0.9, 2.0)	1.6 (0.5, 5.5)	1.4 (0.8, 2.2)
Take-Out Food (*n* = 663)	Crude (excluding missing income)	1.00	1.2 (0.8, 1.8)	0.7 (0.2, 2.7)	2.1 (1.3, 3.3) **
	Adjusted (gender + grade + income)	1.00	1.2 (0.8, 1.8)	0.7 (0.2, 2.7)	1.8 (1.1, 2.9) *

* *p* < 0.05, ** *p* < 0.01.

## Data Availability

The data are not publicly available due to privacy of participants.

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
