# Peer review of "Convenience Food Options and Adequacy of Nutrient Intake among School Children during the COVID-19 Pandemic"

_nutrients, 2022, doi:10.3390/nu14030630_

Round 1

Reviewer 1 Report

The paper focus on the convenience food options and adequacy of nutrient intake among school children during the COVID-19 pandemic. The authors examined the relationship between the frequency of consumption of convenience food options and nutritional status of the school children. They found that frequent consumption of instant or take-out foods among school children result in non-adequate nutritional intake. But various essential analyses were still needed to provide as a supportive proof to claim their findings. I have recommended major revision for publication based on these queries.

  1. Nowadays, there are various convenience food for people, including some healthy and nutritious food. Did you assess the relationship among food preference, food options and adequacy of nutrient intake? In other words, was the balance of food intake a critical factor influencing the association between convenience food options and adequacy of nutrient Intake among school children?
  2. In the second sampling stage, why did you choose the children in grade 5 or grade 8? More details are needed in the Materials and Methods.
  3. In the survey, how did you estimate the data quality of the questionnaire? And what is the criterial of a questionnaire with valid data?
  4. In the Table 1, the classification of “Household annual income” is easy to cause ambiguity for readers.
  5. Many researchers found that person with high BMI showed preference to instant and take-out food. But in your results, no significant difference was detected. This phenomenon should be discussed.
  6. In the Table2, did you exclude confounding factors, such as income, grade or education level of father? More analyses are needed.
  7. Normally, people are more likely to pay attention to the outcomes of high frequent consumption of instant food or non-adequate nutritional intake, such as obesity, fatty liber and diabetes. Although this information was mentioned in the discussion, it would be more meaningful to focus on the relationships between adverse outcomes and food options or nutrient intake, providing some evidence for the formation of health policy.

Author Response

Dear Prof. Aria Chen,

Thank you for the careful and comprehensive review of our manuscript, entitled “Convenience Food Options and Adequacy of Nutrient Intake among School Children during the COVID-19 Pandemic,” for publication. We are grateful for the helpful comments made by our reviewers and extend our thanks to them for improving upon the quality of this manuscript. We have revised our manuscript and provided our point-by-point responses to the reviewer’s comments and recommendations below. The reviewers’ comments are in bold and our responses are in standard typeface. Our manuscript has also been read over by an author who is a native English speaker to correct any grammatical errors. Changes made in the manuscript have been marked using the “Track Changes” function in MS Word.

Sincerely,

Kazue Ishitsuka

Reviewer 1 

Nowadays, there are various convenience food for people, including some healthy and nutritious food. Did you assess the relationship among food preference, food options and adequacy of nutrient intake? In other words, was the balance of food intake a critical factor influencing the association between convenience food options and adequacy of nutrient Intake among school children?

(Our response) Thank you for pointing out this area of improvement in our original analysis. We do not have information about food preference, but we have information about food literacy. Therefore, we have added the results of food literacy and convenience food consumption in our new Table 2, and we added supplemental analyses on food literacy in Table A2 within our appendix. We have also updated the Methods section accordingly.

L155-163 (Method)

“Additionally, the survey we administered included seven questions related to meal preparation literacy. Three questions assessed the guardian’s ability (skill) to prepare meals for their children, two questions assessed their knowledge on what a healthy diet is, and two questions assessed their attitudes towards providing healthy meals to their children. Each response was rated on a Likert scale of 1 to 5. Then, scores were calculated for each of the literacy subgroups (skill, knowledge, and attitude) using the appropriate responses. The maximum score for each subgroup is as follows: 15 for knowledge (3 questions, maximum score of 5 per question), and 10 for skill and attitude (2 questions each, maximum score of 5 per question).”

L188-194 (Results)

“Table 2 lists the means and standard deviations of the skill, knowledge, and attitude literacy scores of the participants based on frequency of convenience food consumption. For those with instant food intake information, there were significant differences in the skill and attitude scores between frequent and seldom consumers. For those with frozen food intake information, there was a significant difference in skill scores between frequent and seldom consumers. Finally, for those with take-out food intake information, there was a significant difference in the attitude scores between frequent and seldom consumers.”

In the second sampling stage, why did you choose the children in grade 5 or grade 8? More details are needed in the Materials and Methods.

(Our response) Thank you for this comment. We have now clarified this choice in our materials and methods section and included a new reference from the Japanese Bureau of Social Welfare and Public Health.

L87-88

“Grades 5 and 8 were chosen as representative of higher elementary school and middle school with reference to previous studies on adolescents in Japan [11].”

In the survey, how did you estimate the data quality of the questionnaire? And what is the criterial of a questionnaire with valid data?

(Our response) Thank you for this comment. Clinically or biologically implausible data was excluded. To assess underreporting and over reporting, school children were excluded when their energy intake below a half of their respective estimated energy requirement or above 1.5 times estimated energy. We took this into account and reran all our analyses after restricting the sample size based on plausible energy intake.

L91-95

“Using the reported energy intakes of the 765 respondents, we defined a restricted range of plausible energy intakes to assess underreporting and overreporting. School children were excluded when their reported energy intake was below half of their respective estimated energy requirement (EER), or above 1.5 times their respective EER [12]. Excluding the outliers resulted in a final sample size of 671 participants.

References

  1. Ministry of Health, Labour and Welfare. Dietary Reference Intake for Japanese (2015). 2018. Available online: https://www.mhlw.go.jp/file/06-Seisakujouhou-10900000-Kenkoukyoku/Full_DRIs2015.pdf

In the Table 1, the classification of “Household annual income” is easy to cause ambiguity for readers.

(Our response) Thank you for this comment. We have reconsidered how to label that row in our Table 1 and have changed the classification scheme. We hope that this is now clearer for our readers.

Many researchers found that person with high BMI showed preference to instant and take-out food. But in your results, no significant difference was detected. This phenomenon should be discussed.

(Our response) Thank you for this comment. We searched published literature, however while we were able to find studies on higher consumption of ultra-processed food and fast-foods related to higher BMI, we could not find much evidence on relationship between instant foods or take-out foods and BMI. We added a new paragraph to our discussion section touching on this point, referencing on existing studies.

L337-353

“While BMI was higher in the frequent consumer group across all three convenience food groups, we failed to observe significant differences between frequent and seldom consumers. There might be a few reasons where we did not observe this result in our study. Other studies on the consumption of ultra-processed food showed large differences in BMI/obesity [28]. While this association may be present in our study, we may not have had the power to observe it. This could also potentially be explained by the school lunch program in Japan, as it was designed to provide Japanese school children with identical, healthy meals, and has been proven to decrease the percentage of overweight and obesity among boys [29]. During the time of our study, school children were attending school, therefore having access to these school lunches. These school lunches provide some degree of a preventive effect against overweight and obesity, therefore making BMI a less sensitive measure in our study. On the other hand, our results are consistent with studies on Korean adults and children and their respective instant noodle consumption [7, 19]. While instant and frozen foods are examples of ultra-processed foods, these results suggest that the correlation between the intake of convenience foods and obesity maybe weaker than that of ultra-processed foods, generally, and obesity. More studies are needed to better understand the association between convenience foods and BMI.”

In the Table 2, did you exclude confounding factors, such as income, grade, or education level of father? More analyses are needed.

(Our response) Thank you for this comment. We agreed with the reviewer’s comment and have since added a second column of p-values to that table (now titled Table 3) which show the p-values derived from multivariate regression on the confounding factors of gender, grade, income. Moreover, our appendix now includes more regression models for mother’s education, and the two significant food literacy scores.

Normally, people are more likely to pay attention to the outcomes of high frequent consumption of instant food or non-adequate nutritional intake, such as obesity, fatty liver and diabetes. Although this information was mentioned in the discussion, it would be more meaningful to focus on the relationships between adverse outcomes and food options or nutrient intake, providing some evidence for the formation of health policy.

(Our response) Thank you for this comment. While we compared BMI between consumers and non-consumers of convenience food in Table 1, our main focus of the study was nutritional intake related with intake of convenience food rather than long-term outcomes of intake, for which a longitudinal study is better suited than a cross-sectional study such as ours. However, we understand the value of focusing on the possible association between intake of convenience food and health outcomes related to non-adequate nutritional intake, thus we added more to our discussion section accordingly.

L337-353

“While BMI was higher in the frequent consumer group across all three convenience food groups, we failed to observe significant differences between frequent and seldom consumers. There might be a few reasons where we did not observe this result in our study. Other studies on the consumption of ultra-processed food showed large differences in BMI/obesity [28]. While this may be present in our study, we may not have had the power to observe it. This could also potentially be explained by the school lunch program in Japan, as it was designed to provide Japanese schoolchildren with identical, healthy meals, and has been proven to decrease the percentage of overweight and obesity among boys [29]. During the time of our study, schoolchildren were attending school, therefore having access to these school lunches. These school lunches provide some degree of a preventive effect against overweight and obesity, therefore making BMI a less sensitive measure in our study. On the other hand, our results are consistent with studies on Korean adults and children and their respective instant noodle consumption [7, 19]. While instant and frozen foods are examples of ultra-processed foods, these results suggest that the correlation between the intake of convenience foods and obesity maybe weaker than that of ultra-processed foods, generally, and obesity. More studies are needed to better understand the association between convenience foods and BMI.”

Reviewer 2

Keywords: In my opinion, it would be better to use the word "convenience food" in keywords than "instant food".

(Our response) Thank you for this comment. We have replaced “instant food” with “convenience food” in our keywords.

Introduction

Can you explain why respondents chose convenience food during the pandemic?

(Our response) Thank you for the question. We chose convenience foods during the pandemic because research suggests an increase in consumption of convenience foods since the beginning of the COVID-19 pandemic [5].

In the introduction, I miss the context of how the Japanese eat habitually. This is important for the international context of the article for readers from other countries who are unfamiliar with the Japanese specifics. As far as I know, processed foods and gastronomic foods are used quite often in Japan.

(Our response) Thank you for these comments. We have added a section in our introduction now about the habitual consumption of processed foods in Japan, and how food environments have since changed due to COVID.

L43-50

“Before the pandemic, Japan already had one of the higher country-wide retail sales of ultra-processed foods (such as packaged snacks or prepared frozen dishes), which is a strong indicator of unfavorable nutrient intake [4]. Once the pandemic began, countries around the world saw an increase in purchases and consumption of convenience foods, such as instant and frozen foods [5]. Additionally, in Japan, those with worsened household income self-reported less time for cooking and higher rates of consuming take-out foods [6]. Therefore, these changing household food environments are expected to have a direct impact on the nutritional intake of schoolchildren.”

Material and Methods

I don’t understand the sentence “Those with an energy intake of 0 were excluded from the final analysis, resulting in a final sample size of 762” (lines 79-80). What did the authors mean? There were 1,500 households in the group of respondents, is it possible that over 700 respondents indicated that they do not eat food or do not eat convenience food? This requires some explanation.

Thank you very much for this comment. We recognized how this was unclear. We altered the criteria for inclusion based on comments on other reviewers. Together, we have now clarified our methodology in this section.

L91-95

“Out of 1500 households administered the BDHQ, 765 (51.0%) completed the survey. Using the reported energy intakes of the 765 respondents, we defined a restricted range of plausible energy intakes to assess underreporting and overreporting. Schoolchildren were excluded when their reported energy intake was below half of their respective estimated energy requirement (EER), or above 1.5 times their respective EER. Excluding the outliers resulted in a final sample size of 671 participants.”

Limitation

I have a question. Is there any limitation to these results? If yes, it is worth writing about it.

(Our response) Thank you for this comment. We have added more to our discussion section regarding the limitation of these results.

L380-395

Firstly, a moderate response rate of 51.0% may limit the generalizability of our findings. Nevertheless, the characteristics of the schoolchildren in our study are comparable to those in the Japanese national survey. Moreover, household income in the present study was comparable to those among households with children in the Japanese national survey (24, 38). Secondly, inaccuracies due to BDHQ self-reporting, such as the potential for over- or underestimating. While the BDHQ serves as a useful tool for diet recall over a longer period, the additional use of measurements biomarkers would strengthen the analysis of such a study. Thirdly, this study cannot be generalized to an older population or a non-school attending population, due to the inclusion of only grade-school aged children, and the implications of schoolchildren having access to healthy school lunches that non- schoolchildren would lack. Finally, due to the cross-sectional nature of our study, our analyses were limited to reporting nutritional intake by convenience food option and we were not able assess its long-term health effects. Future studies assessing the resulting potential for adverse health effects of altered nutritional intake due to increased convenience food intake are needed.

Conclusion

What are the practical and theoretical implications of the research? The current conclusions are quite enigmatic.

(Our response) Thank you very much for these comments. We have revised our conclusion with these points in mind.

L409-415

“The evidence from our study suggests that frequent instant and take-out food consumption are associated with non-adequate nutritional intake. With the COVID-19 pandemic ongoing, there is likely to be even more children dependent on convenience food options in the near future. Providing convenience food options that are more nutrient rich, as well as recommending families to choose healthier items when consuming convenience food options, may help to prevent long-term adverse health effects and are needed for children for whom reduction of intake of convenience food options itself is not possible.”

References

Many references (11 out of 33) come from the last 3 years. References are cited according to journal rules.

Thank you very much for reviewing our references.

Reviewer 2 Report

The manuscript (nutrients-1529421) submitted for review is very interesting.

Authors, Please note and address the following comments:

Keywords: In my opinion, it would be better to use the word "convenience food" in keywords than "instant food".

Introduction

Can you explain why respondents chose convenience food during the pandemic?

In the introduction, I miss the context of how the Japanese eat habitually. This is important for the international context of the article for readers from other countries who are unfamiliar with the Japanese specifics. As far as I know, processed foods and gastronomic foods are used quite often in Japan.

Material and Methods: I don’t understand the sentence “Those with an energy intake of 0 were excluded from the final analysis, resulting in a final sample size of 762” (lines 79-80). What did the authors mean? There were 1,500 households in the group of respondents, is it possible that over 700 respondents indicated that they do not eat food or do not eat convenience food? This requires some explanation.

Limitation

I have a question. Is there any limitation to these results? If yes, it is worth writing about it.

Conclusion

What are the practical and theoretical implications of the research?

The current conclusions are quite enigmatic.

References

Many references (11 out of 33) come from the last 3 years. References are cited according to journal rules.

Reviewer

Author Response

Dear Prof. Aria Chen,

Thank you for the careful and comprehensive review of our manuscript, entitled “Convenience Food Options and Adequacy of Nutrient Intake among School Children during the COVID-19 Pandemic,” for publication. We are grateful for the helpful comments made by our reviewers and extend our thanks to them for improving upon the quality of this manuscript. We have revised our manuscript and provided our point-by-point responses to the reviewer’s comments and recommendations below. The reviewers’ comments are in bold and our responses are in standard typeface. Our manuscript has also been read over by an author who is a native English speaker to correct any grammatical errors. Changes made in the manuscript have been marked using the “Track Changes” function in MS Word.

Sincerely,

Kazue Ishitsuka

Reviewer 1

Nowadays, there are various convenience food for people, including some healthy and nutritious food. Did you assess the relationship among food preference, food options and adequacy of nutrient intake? In other words, was the balance of food intake a critical factor influencing the association between convenience food options and adequacy of nutrient Intake among school children?

(Our response) Thank you for pointing out this area of improvement in our original analysis. We do not have information about food preference, but we have information about food literacy. Therefore, we have added the results of food literacy and convenience food consumption in our new Table 2, and we added supplemental analyses on food literacy in Table A2 within our appendix. We have also updated the Methods section accordingly.

L155-163 (Method)

“Additionally, the survey we administered included seven questions related to meal preparation literacy. Three questions assessed the guardian’s ability (skill) to prepare meals for their children, two questions assessed their knowledge on what a healthy diet is, and two questions assessed their attitudes towards providing healthy meals to their children. Each response was rated on a Likert scale of 1 to 5. Then, scores were calculated for each of the literacy subgroups (skill, knowledge, and attitude) using the appropriate responses. The maximum score for each subgroup is as follows: 15 for knowledge (3 questions, maximum score of 5 per question), and 10 for skill and attitude (2 questions each, maximum score of 5 per question).”

L188-194 (Results)

“Table 2 lists the means and standard deviations of the skill, knowledge, and attitude literacy scores of the participants based on frequency of convenience food consumption. For those with instant food intake information, there were significant differences in the skill and attitude scores between frequent and seldom consumers. For those with frozen food intake information, there was a significant difference in skill scores between frequent and seldom consumers. Finally, for those with take-out food intake information, there was a significant difference in the attitude scores between frequent and seldom consumers.”

In the second sampling stage, why did you choose the children in grade 5 or grade 8? More details are needed in the Materials and Methods.

(Our response) Thank you for this comment. We have now clarified this choice in our materials and methods section and included a new reference from the Japanese Bureau of Social Welfare and Public Health.

L87-88

“Grades 5 and 8 were chosen as representative of higher elementary school and middle school with reference to previous studies on adolescents in Japan [11].”

In the survey, how did you estimate the data quality of the questionnaire? And what is the criterial of a questionnaire with valid data?

(Our response) Thank you for this comment. Clinically or biologically implausible data was excluded. To assess underreporting and over reporting, school children were excluded when their energy intake below a half of their respective estimated energy requirement or above 1.5 times estimated energy. We took this into account and reran all our analyses after restricting the sample size based on plausible energy intake.

L91-95

“Using the reported energy intakes of the 765 respondents, we defined a restricted range of plausible energy intakes to assess underreporting and overreporting. School children were excluded when their reported energy intake was below half of their respective estimated energy requirement (EER), or above 1.5 times their respective EER [12]. Excluding the outliers resulted in a final sample size of 671 participants.”

References

  1. Ministry of Health, Labour and Welfare. Dietary Reference Intake for Japanese (2015). 2018. Available online: https://www.mhlw.go.jp/file/06-Seisakujouhou-10900000-Kenkoukyoku/Full_DRIs2015.pdf

In the Table 1, the classification of “Household annual income” is easy to cause ambiguity for readers.

(Our response) Thank you for this comment. We have reconsidered how to label that row in our Table 1 and have changed the classification scheme. We hope that this is now clearer for our readers.

Many researchers found that person with high BMI showed preference to instant and take-out food. But in your results, no significant difference was detected. This phenomenon should be discussed.

(Our response) Thank you for this comment. We searched published literature, however while we were able to find studies on higher consumption of ultra-processed food and fast-foods related to higher BMI, we could not find much evidence on relationship between instant foods or take-out foods and BMI. We added a new paragraph to our discussion section touching on this point, referencing on existing studies.

L337-353

“While BMI was higher in the frequent consumer group across all three convenience food groups, we failed to observe significant differences between frequent and seldom consumers. There might be a few reasons where we did not observe this result in our study. Other studies on the consumption of ultra-processed food showed large differences in BMI/obesity [28]. While this association may be present in our study, we may not have had the power to observe it. This could also potentially be explained by the school lunch program in Japan, as it was designed to provide Japanese school children with identical, healthy meals, and has been proven to decrease the percentage of overweight and obesity among boys [29]. During the time of our study, school children were attending school, therefore having access to these school lunches. These school lunches provide some degree of a preventive effect against overweight and obesity, therefore making BMI a less sensitive measure in our study. On the other hand, our results are consistent with studies on Korean adults and children and their respective instant noodle consumption [7, 19]. While instant and frozen foods are examples of ultra-processed foods, these results suggest that the correlation between the intake of convenience foods and obesity maybe weaker than that of ultra-processed foods, generally, and obesity. More studies are needed to better understand the association between convenience foods and BMI.”

In the Table 2, did you exclude confounding factors, such as income, grade, or education level of father? More analyses are needed.

(Our response) Thank you for this comment. We agreed with the reviewer’s comment and have since added a second column of p-values to that table (now titled Table 3) which show the p-values derived from multivariate regression on the confounding factors of gender, grade, income. Moreover, our appendix now includes more regression models for mother’s education, and the two significant food literacy scores.

Normally, people are more likely to pay attention to the outcomes of high frequent consumption of instant food or non-adequate nutritional intake, such as obesity, fatty liver and diabetes. Although this information was mentioned in the discussion, it would be more meaningful to focus on the relationships between adverse outcomes and food options or nutrient intake, providing some evidence for the formation of health policy.

(Our response) Thank you for this comment. While we compared BMI between consumers and non-consumers of convenience food in Table 1, our main focus of the study was nutritional intake related with intake of convenience food rather than long-term outcomes of intake, for which a longitudinal study is better suited than a cross-sectional study such as ours. However, we understand the value of focusing on the possible association between intake of convenience food and health outcomes related to non-adequate nutritional intake, thus we added more to our discussion section accordingly.

L337-353

“While BMI was higher in the frequent consumer group across all three convenience food groups, we failed to observe significant differences between frequent and seldom consumers. There might be a few reasons where we did not observe this result in our study. Other studies on the consumption of ultra-processed food showed large differences in BMI/obesity [28]. While this may be present in our study, we may not have had the power to observe it. This could also potentially be explained by the school lunch program in Japan, as it was designed to provide Japanese schoolchildren with identical, healthy meals, and has been proven to decrease the percentage of overweight and obesity among boys [29]. During the time of our study, schoolchildren were attending school, therefore having access to these school lunches. These school lunches provide some degree of a preventive effect against overweight and obesity, therefore making BMI a less sensitive measure in our study. On the other hand, our results are consistent with studies on Korean adults and children and their respective instant noodle consumption [7, 19]. While instant and frozen foods are examples of ultra-processed foods, these results suggest that the correlation between the intake of convenience foods and obesity maybe weaker than that of ultra-processed foods, generally, and obesity. More studies are needed to better understand the association between convenience foods and BMI.”

Reviewer 2

Keywords: In my opinion, it would be better to use the word "convenience food" in keywords than "instant food".

(Our response) Thank you for this comment. We have replaced “instant food” with “convenience food” in our keywords.

Introduction

Can you explain why respondents chose convenience food during the pandemic?

(Our response) Thank you for the question. We chose convenience foods during the pandemic because research suggests an increase in consumption of convenience foods since the beginning of the COVID-19 pandemic [5].

In the introduction, I miss the context of how the Japanese eat habitually. This is important for the international context of the article for readers from other countries who are unfamiliar with the Japanese specifics. As far as I know, processed foods and gastronomic foods are used quite often in Japan.

(Our response) Thank you for these comments. We have added a section in our introduction now about the habitual consumption of processed foods in Japan, and how food environments have since changed due to COVID.

L43-50

“Before the pandemic, Japan already had one of the higher country-wide retail sales of ultra-processed foods (such as packaged snacks or prepared frozen dishes), which is a strong indicator of unfavorable nutrient intake [4]. Once the pandemic began, countries around the world saw an increase in purchases and consumption of convenience foods, such as instant and frozen foods [5]. Additionally, in Japan, those with worsened household income self-reported less time for cooking and higher rates of consuming take-out foods [6]. Therefore, these changing household food environments are expected to have a direct impact on the nutritional intake of schoolchildren.”

Material and Methods

I don’t understand the sentence “Those with an energy intake of 0 were excluded from the final analysis, resulting in a final sample size of 762” (lines 79-80). What did the authors mean? There were 1,500 households in the group of respondents, is it possible that over 700 respondents indicated that they do not eat food or do not eat convenience food? This requires some explanation.

Thank you very much for this comment. We recognized how this was unclear. We altered the criteria for inclusion based on comments on other reviewers. Together, we have now clarified our methodology in this section.

L91-95

“Out of 1500 households administered the BDHQ, 765 (51.0%) completed the survey. Using the reported energy intakes of the 765 respondents, we defined a restricted range of plausible energy intakes to assess underreporting and overreporting. Schoolchildren were excluded when their reported energy intake was below half of their respective estimated energy requirement (EER), or above 1.5 times their respective EER. Excluding the outliers resulted in a final sample size of 671 participants.”

Limitation

I have a question. Is there any limitation to these results? If yes, it is worth writing about it.

(Our response) Thank you for this comment. We have added more to our discussion section regarding the limitation of these results.

L380-395

Firstly, a moderate response rate of 51.0% may limit the generalizability of our findings. Nevertheless, the characteristics of the schoolchildren in our study are comparable to those in the Japanese national survey. Moreover, household income in the present study was comparable to those among households with children in the Japanese national survey (24, 38). Secondly, inaccuracies due to BDHQ self-reporting, such as the potential for over- or underestimating. While the BDHQ serves as a useful tool for diet recall over a longer period, the additional use of measurements biomarkers would strengthen the analysis of such a study. Thirdly, this study cannot be generalized to an older population or a non-school attending population, due to the inclusion of only grade-school aged children, and the implications of schoolchildren having access to healthy school lunches that non- schoolchildren would lack. Finally, due to the cross-sectional nature of our study, our analyses were limited to reporting nutritional intake by convenience food option and we were not able assess its long-term health effects. Future studies assessing the resulting potential for adverse health effects of altered nutritional intake due to increased convenience food intake are needed.

Conclusion

What are the practical and theoretical implications of the research? The current conclusions are quite enigmatic.

(Our response) Thank you very much for these comments. We have revised our conclusion with these points in mind.

L409-415

“The evidence from our study suggests that frequent instant and take-out food consumption are associated with non-adequate nutritional intake. With the COVID-19 pandemic ongoing, there is likely to be even more children dependent on convenience food options in the near future. Providing convenience food options that are more nutrient rich, as well as recommending families to choose healthier items when consuming convenience food options, may help to prevent long-term adverse health effects and are needed for children for whom reduction of intake of convenience food options itself is not possible.”

References

Many references (11 out of 33) come from the last 3 years. References are cited according to journal rules.

Thank you very much for reviewing our references.

Round 2

Reviewer 1 Report

Thanks very much for your revision. Based on your manuscript, I have another question: In Table 2, how did you rate the responses about these three food literacies? What was the criterial?

Author Response

Thank you for the careful and comprehensive review of our manuscript, entitled “Convenience Food Options and Adequacy of Nutrient Intake among School Children during the COVID-19 Pandemic,” for publication. We are grateful for the helpful comments made by our reviewers and extend our thanks to them for improving upon the quality of this manuscript. We have revised our manuscript and provided our point-by-point responses to the reviewer’s comments and recommendations below. The reviewers’ comments are in bold and our responses are in standard typeface. We have also corrected several grammatical errors. Changes made in the manuscript have been marked using the “Track Changes” function in MS Word.

Sincerely,

Kazue Ishitsuka

Reviewer 1

 Thanks very much for your revision. Based on your manuscript, I have another question: In Table 2, how did you rate the responses about these three food literacies? What was the criterial?

(Our response) Thank you for your comments. We have added the criteria for how the authors rated the responses about these three food literacies.

Line 149-154

Each response was rated with a 5-point Likert scale (1, yes/always; 2, yes/sometimes; 3, unsure; 4, no/ not so much; and 5, no/not at all, for skill and attitude questions: 1, not at all; 2, understand a little; 3, neither; 4, understand mostly; 5, fully understand for knowledge questions). These variables were treated as numerical, and the scores for attitude and affordability questions were reversed. Then, the final score for each food literacy group was obtained from adding their respective questions.